# Peptide Antibody Reactivity to Homologous Regions in Glutamate Decarboxylase Isoforms and Coxsackievirus B4 P2C

**DOI:** 10.3390/ijms23084424

**Published:** 2022-04-17

**Authors:** Nicole Hartwig Trier, Niccolo Valdarnini, Ilaria Fanelli, Paolo Rovero, Paul Robert Hansen, Claus Schafer-Nielsen, Evaldas Ciplys, Rimantas Slibinskas, Flemming Pociot, Tina Friis, Gunnar Houen

**Affiliations:** 1Department of Neurology, Rigshospitalet Glostrup, Valdemar Hansens vej 13, 2600 Glostrup, Denmark; 2Interdepartmental Laboratory of Peptide and Protein Chemistry and Biology, Department of NeuroFarBa, University of Florence, Via Ugo Schiff 6, I-50019 Sesto Fiorentino, Italy; valdarnini91@gmail.com (N.V.); ilaria.fanelli1@stud.unifi.it (I.F.); paolo.rovero@unifi.it (P.R.); 3Department of Drug Design and Pharmacology, University of Copenhagen, Universitetsparken 2, 2100 Copenhagen, Denmark; prh@sund.ku.dk; 4Schafer-N, Lersø Parkallé 42, 2100 Copenhagen, Denmark; peptides@schafer-n.com; 5Life Sciences Center, Institute of Biotechnology, Vilnius University, Saulėtekio al. 7, LT-10257 Vilnius, Lithuania; evaldas.ciplys@bti.vu.lt (E.C.); rimantas.slibinskas@bti.vu.lt (R.S.); 6Steno Diabetes Center, Borgmester Ib Juuls Vej 83, 2730 Hellerup, Denmark; flemming.pociot@regionh.dk; 7Department of Autoimmunity and Biomarkers, Statens Serum Institut, Artillerivej 5, 2300 Copenhagen, Denmark; tfs@ssi.dk; 8Department Biochemistry and Molecular Biology, University of Southern Denmark, Campusvej 55, 5230 Odense, Denmark

**Keywords:** antibody cross-reactivity, coxsackievirus, glutamate decarboxylase, peptide antibodies, stiff-person syndrome, type 1 diabetes

## Abstract

Two isoforms of the glutamate decarboxylase (GAD) enzyme exist, GAD65 and GAD67, which are associated with type 1 diabetes (T1D) and stiff-person syndrome (SPS), respectively. Interestingly, it has been reported that T1D patients seldom develop SPS, whereas patients with SPS occasionally develop T1D. In addition, coxsackievirus B4 (CVB4) has previously been proposed to be involved in the onset of T1D through molecular mimicry. On this basis, we aimed to examine antibody cross-reactivity between a specific region of GAD65 and GAD67, which has high sequence homology to the nonstructural P2C protein of CVB4 to determine potential correlations at antibody level. Monoclonal peptide antibodies generated in mice specific for a region with high similarity in all three proteins were screened for reactivity along with human sera in immunoassays. In total, six antibodies were generated. Two of the antibodies reacted to both GAD isoforms. However, none of the antibodies were cross-reactive to CVB, suggesting that antibody cross-reactivity between GAD65 and CVB, and GAD67 and CVB may not contribute to the onset of T1D and SPS, respectively.

## 1. Introduction

Type-1 diabetes (T1D) is a chronic disease that requires treatment throughout life. T1D accounts for 5–10% of diabetes cases worldwide, affecting men as well as women [1,2]. The risk of T1D development in the general population is 1:300, and the annual incidence is increasing by 2.3% per year [2,3,4]. Especially children in the early years (0–4) are in the risk zone of developing T1D, with a doubling in the number of affected children over approximately 12 years [3,5]. Characteristic for T1D is that the insulin-producing pancreatic β-cells slowly are degraded by CD4 and CD8 T cells and macrophages infiltrating the islets, which ultimately leads to progressive β cell loss [6,7]. Destruction of β cells results in insulin insufficiency, and if left untreated, patients develop life-threatening hyperglycemia that clinically manifests with weight loss, polyuria, and polydipsia [8]. Consequently, patients with T1D remain insulin-dependent for their lifespan [6]. In addition to clinical symptoms, serologic biomarkers are specific for T1D patients, among others, antibodies to glutamic acid decarboxylase (GAD), which often are present prior to the clinical diagnosis [9]. GAD is a pyridoxal phosphate (PLP)-dependent enzyme that catalyses the conversion of L-glutamic acid to the inhibitory neurotransmitter γ-aminobutyric acid (GABA) and is expressed in GABA-secreting neurons and in pancreatic β cells [10,11,12].

Besides being associated with T1D, GAD antibodies are detected in individuals suffering from Stiff Person Syndrome (SPS) [10,13,14,15]. SPS is a neurologic disease that is characterized by fluctuating muscle rigidity in the trunk and limbs and a heightened sensitivity to stimuli such as noise, touch, and emotional distress, which can induce muscle spasms [16,17]. SPS affects twice as many women as men, primarily in the age of 30–50. Currently, the etiology of SPS remains unknown.

SPS is frequently associated with T1D. An increased rate of disease risk has been found to be related to antibody cross-reactivity between the two diseases, as two isoforms of GAD exist, GAD65 and GAD67 [10,18]. The two GAD isoforms differ in their enzyme activity and localization. GAD67 is primarily located in the cytoplasm, is constitutively active, and provides for the steady basal production of GABA, whereas GAD65 mainly is present in synaptic vesicles, undergoes auto-inactivation during enzyme activity, and occurs in the cell, providing for a pulse in production under circumstances that demand a rapid surge of GABA synthesis and release [14,18,19]. The two isoforms show high sequence similarity, with the middle and C-terminal domains having 74% identity but differing (25% identity) in the N-terminal domain, mainly in the first 100 amino acids [10,13,14]. Due to this high degree of similarity antibody, cross-reactivity prevails between the two isoforms [20].

It has been demonstrated that GAD65 and not GAD67 is the major autoantigen in T1D, although antibodies to GAD67 occasionally are associated with T1D as well [18,21]. This may be due to the low expression levels of GAD67 in the pancreas. In contrast, antibodies to GAD67 are associated with SPS, which makes sense as SPS is a neurologic disease [14]. However, it has been proposed that individuals with SPS have a higher tendency to develop T1D, which may be related to differences in epitope reactivity patterns and cross-reactivity between GAD isoforms; however, the exact reason remains to be determined.

Viruses such as coxsackievirus (CV) have been suggested to be associated with T1D onset [22,23,24,25,26,27,28,29,30,31,32]. CVs are of the RNA enterovirus family. Coxsackieviruses B (CVB) in particular have been suggested to be involved in the onset of T1D [22,28,30,31,32,33]. CVB4 was one of the first enteroviruses to be associated with T1D. Enteroviruses infect β cells, and thus pancreatic β cells and not α cells are chronically infected [25,27]. CVB has been proposed to induce onset of T1D through molecular mimicry, as a small region of the nonstructural P2C protein in CVB4 shows sequence homology to the GAD isoforms [33] (Figure 1).

GAD65 contains a 6-amino acid sequence PEVKEK, which is found in er nonstructural P2C of CVB as well. Besides this sequence similarity in position 260–265 and 38–42 of GAD65 and CVB P2C, respectively, sequence homology between GAD65, P2C, and GAD67 exists, as GAD67 contains the ^268^PEVKTK^273^ motif. On the basis of this knowledge, we generated antibodies to peptides containing the homologue region originating from GAD65, GAD67, and CVB P2C to determine the potential role of molecular mimicry as a contributor of T1D and SPS onset.

## 2. Results

### 2.1. Test of Mouse Bleeds in Enzyme-Linked Immunosorbent Assay

In total, 12 mice were immunized with peptides conjugated to keyhole limpet hemocyanin (KLH). Mice from groups A, B, and C were immunized with GAD67, GAD65, and CVB, respectively, whereafter bleeds were tested for antibody reactivity in an enzyme-linked immunosorbent assay (ELISA) (Figure 2), where bound antibodies are determined by a colorimetric reaction.

Bleeds from mice of group A and B mainly reacted with GAD65 and GAD67 with a few exceptions. Only limited reactivity to CVB was detected, although two mice from group B reacted to GAD67 after repeated injections (4–5). Mice from group C immunized with CVB reacted with all three peptides, most pronounced to GAD65 and CVB.

Collectively, the polyclonal antibody response was very cross-reactive between the two GAD isoforms, which relates to the high degree of sequence homology. Similarly, limited antibody reactivity was found to CVB, which related to the limited sequence homology to CVB.

On the basis of the current results, mice with high antibody titers and high degree of cross-reactive antibody responses were selected for further antibody production.

### 2.2. Antibody Reactivity to Coxsackievirus and Glutamic Acid Decarboxylase Peptides

Following immunization and final cloning and screening, six stable clones were selected for further analysis (Table 1). Moreover, as peptide presentation in immunoassays is dependent on type of coating, the selected antibodies were tested in different assay setups.

First, the antibodies were tested in traditional ELISA, where the free peptides were coated directly to the solid surface of microtiter plates.

SSI-HYB 386-01 mainly recognized GAD65 and weakly GAD67, SSI-HYB 386-02 recognized GAD65 and GAD67, and SSI-HYB 387-01 and SSI-HYB 387-02 recognized GAD67. SSI-HYB 389-01 and -02 only recognized GAD65 (Figure 3a). None of the antibodies recognized the CVB peptide.

The antibodies in general recognized C-terminal biotinylated peptides in a pattern similar to the free peptides (Figure 3b). However, when comparing antibody reactivity to N-terminal biotinylated peptides, significant differences appeared, as only the SSI-HYB 387-01 recognized the C-terminal biotinylated peptide. The remaining antibodies did not recognize the C-terminal biotinylated peptides.

These findings were confirmed when testing antibody reactivity to the resin-bound peptides, where the peptides are attached to the resin beads through their C-terminal end. As seen, only SSI-HYB 387-01 reacted with the resin-bound peptides (Figure 3c), indicating that the antibodies generated had a dependency of a free C-terminal.

### 2.3. Epitope Characterization of SSI-HYB 387-01

SSI-HYB 387-01 recognized both N- and C-terminal-presented peptides, indicating that the epitope is located in the center of the GAD67 peptide. To identify the binding site of SSI-HYB 387-01, antibody reactivity to N- and C-terminal-truncated resin-bound peptides was tested by modified ELISA.

Screening of N-terminal-truncated peptides revealed that Arg^6^ was essential for antibody reactivity, as peptides without Arg^6^ were not recognized by SSI-HYB 387-01 (Figure 4a). Screening of C-terminal-truncated peptides identified Pro^11^ as essential for antibody reactivity, as antibody reactivity to peptide SIMAARYKYF missing the C-terminal Pro^11^ was reduced by 60%. Absence of Phe^10^ reduced antibody reactivity by 100% compared to the GAD67 control peptide.

On the basis of the current findings, the sequence ^6^RYKYFP^11^ was identified as the epitope of SSI-HYB 387-01, which was confirmed by competitive inhibition assay. Figure 5a illustrates the competitive ability of truncated epitope versions, where microtiter plates were coated with the GAD67 peptide, and the inhibitory effect of the truncated peptides containing the essential regions were determined.

As seen, the peptide RYKYF did not inhibit antibody to the GAD67 control peptide, whereas the peptide RYKYFP inhibited antibody reactivity significantly in comparison to the control, where no inhibitor was added. The remaining peptides, containing the complete epitope, inhibited antibody reactivity as well.

To determine epitope characteristics, Ala scanning and functionality scanning were conducted by competitive inhibition assays (Figure 5b). The sequence ^6^RYKYFP^11^ was used for generation of Ala-substituted and functionality-substituted peptides, where each amino acid was substituted one by one, whereafter the inhibitory effect was determined in microtiter plates coated with the full-length GAD67 peptide.

As seen, not all amino acids in the epitope contributed equally to antibody binding. [Lys→Ala]^3^ substitution reduced antibody reactivity by 100%, whereas [Tyr→Ala]^4^ substitution did not influence antibody reactivity relative to the control, where no inhibitor was added. [Arg→Ala]^1^- and [Phe→Ala]^5^-substituted peptides inhibited antibody reactivity by approximately 20%, whereas [Tyr→Ala]^2^ and [Pro→Ala]^5^ substitutions reduced antibody reactivity by approximately 60%.

Screening of functionality-substituted peptides in general confirmed results of Ala-substituted peptides. [Lys→Arg]^3^ substitution reduced antibody reactivity by 100%, whereas [Tyr→Thr]^4^ and [Tyr→Trp]^5^ did not influence antibody reactivity relative to the control, where no inhibitor was added. [Arg→Lys]^1^ substitution reduced SSI-HYB 387-01 reactivity by approximately 30%, whereas [Tyr→Thr]^2^ substitution reduced antibody reactivity by approximately 80% relative to the control, where no inhibitor was added.

Finally, to determine whether epitope presentation interferes with antibody recognition, relevant amino acids were substituted with D-amino acids and analyzed by competitive inhibition assay. Figure 5c illustrates the reactivity of SSI-HYB 387-01 to the substituted peptides. [K→k]^3^ and [Y→y]^4^ substitutions did not inhibit antibody reactivity, whereas the peptide containing the [R→r]^1^ substitution inhibited antibody reactivity by approximately 95%.

Finally, to determine whether the antibody recognizes the native GAD67 protein, SSI-HYB 387-01 was tested for reactivity to reduced and non-reduced full-length GAD67 protein by immunostaining.

Figure 6a represents the reactivity of SSI-HYB 387-01 to GAD67 in Western blotting. Only the defolded protein was recognized by SSI-HYB 387-01, indicating that the antibody does not recognize the native protein structure or that the native GAD67 was not present as a monomer in the native state, but only as dimers or tetramers. To examine this further, a Coomassie Brilliant Blue staining of GAD67 in SDS-PAGE was conducted, which verified that GAD67 was not found as a monomer (Figure 5b). Thus, SSI-HYB 387-01 only recognized the protein in an unfolded state or when present as a dimer.

Finally, a titration of SSI-HYB 387-01 was conducted to evaluate antibody affinity. As seen (Figure 7), 50% of max binding was observed at a dilution of approximately 1:200.

### 2.4. Characterization of Epitopes to SSI-HYB 386-01, 386-02, 387-01, 389-01, and 389-02

Preliminary findings illustrated that the remaining SSI-HYBs do not recognize peptides presented C-terminally by biotin labelling or resin, indicating that these SSI-HYBs have a dependency for the C-terminal free acid.

To determine the epitopes of the remaining antibodies, HYB reactivity to relevant truncated and substituted peptides was analyzed by competitive immunoassays.

Figure 8 illustrates the reactivity of the cross-reactive SSI-HYB 386-01 and SSI-HYB 386-02 to GAD65 and GAD67 peptides. 

As presented, all peptides inhibited SSI-HYB 386-02 reactivity to GAD65 and GAD67 peptides, indicating that this HYB mainly recognizes peptide backbone in combination with a central free C-terminal -COOH group.

SSI-HYB 386-01 reactivity to truncated and substituted GAD peptides indicated that this HYB is backbone-dependent as well, especially as all peptides inhibited antibody reactivity to GAD67. In contrast, four peptides—GMAALPAL, GMAAVPAL, KTGMAAVPK, and AALPAL—inhibited antibody reactivity by approximately 65, 55, 40, and 65%, respectively, to the GAD65 peptides. In three of the peptides, a central Arg in the C-terminal was substituted with Ala, whereas the KTGMAAVPK peptide was truncated in the C-terminal, missing the amino acid Leu. These findings indicate that primarily the peptide backbone and to some degree the C-terminal amino acids -RL are essential for antibody reactivity, identifying the complete epitope as xxxxRL-COOH. 

As the SSI-HYBs 386-01 and 386-02 recognize both GAD isoforms, a titration experiment was conducted to evaluate the affinities of the antibodies to the GAD peptides.

Figure 9 illustrates the reactivity of SSI-HYB 386-01 and SSI-HYB 386-02 to GAD65 and GAD67. 

As seen, SSI-HYB 386-01 obtained a higher reactivity to GAD65 compared to GAD67, whereas SSI-HYB 386-02 obtained a higher reactivity to GAD67 compared to GAD65. This may be explained by the fact that the GAD65 peptide is the immunogen of SSI-HYB 386-01, whereas the GAD67 peptide is the immunogen of SSI-HYB 386-02. Approximately 50% of SSI-HYB 386-01 reactivity was reduced at a 1:40 dilution for both GAD65 and GAD67, whereas 50% of SSI-HYB 386-02 reactivity to GAD67 and GAD65 was reduced at a dilution of approximately 1:80 and 1:120, respectively, indicating that the affinity was slightly higher to GAD65, as expected.

Next the epitope of SSI-HYB 389-01 and 389-02 were characterized. Figure 10a illustrates the reactivity of SSI-HYB 389-01 and SSI-HYB 389-02 to substituted peptides. As seen, all of the truncated and substituted peptides inhibited antibody reactivity, indicating that these antibodies primarily recognized peptide backbone in combination with a free C-terminal acid. A similar pattern was observed for SSI-HYB 387-02 reactivity (Figure 10b), indicating that this antibody recognized the peptide backbone as well.

To verify the dependency of peptide backbone for HYB reactivity, inhibitory studies were conducted to determine whether random peptides in theory may inhibit the reactivity of SSI-HYB 386-01, 386-02, and 387-02. Wells were coated with GAD65 peptide for SSI-HYB 386-01 and 386-02 and GAD67 for SSI-HYB 387-02, and antibody reactivity was inhibited with peptides from GAD65/67, CVB, and a random control peptide, with no sequence similarity to GAD or CVB (GQGRGRWRGRGRSKGRGRMH).

Figure 11 illustrates the reactivity of SSI-HYB 386-01, 386-02, and 387-02 to GAD65, GAD65, and GAD67, respectively. As seen, a similar pattern was observed for all three cell culture supernatants. The specific peptide in question (GAD65/67) inhibited antibody reactivity to the same level as the CVB peptide and the random control peptide. These results indicate that the antibodies depend on the free C-terminal with a combination of peptide backbone.

### 2.5. Reactivity of Human Sera to Glutamic Acid Decarboxylase and Coxsackievirus Peptides

It has previously been proposed that cross-reactivity between CVB and GAD proteins is a contributor to the onset of T1D through molecular mimicry. On the basis of this hypothesis, we tested the reactivity of a T1D sera and HCs to the three peptides showing sequence homology in ELISA.

Figure 12 illustrates the reactivity of T1D and HD sera to peptides from GAD65, GAD67, and CVB with sequence similarity. No reactivity was found for the three peptides, indicating that cross-reactivity within this region is not a contributor to onset of T1D and that the epitope analyzed is not an immunodominant epitope in T1D, as previously proposed.

## 3. Discussion

Molecular mimicry is a well-accepted theory for initiation of diseases, and structural as well as sequence homology has been proposed as being capable of inducing autoimmune disease onset [34]. In the current study, we aimed to analyze whether antibody cross-reactivity between CVB4 P2C and GAD may contribute to the onset of T1D and/or SPS. In total, 12 mice were immunized with peptides originating from GAD65, GAD67, or CVB P2C, and the immune responses of the majority of mice showed some degree of cross-reactivity. Of special interest were the mice immunized with CVB (mice in group C), which showed cross-reactivity to GAD65 and GAD67 as well. However, no stable monoclonal CVB antibodies or HYBs, which showed cross-reactivity between CVB or GAD, were generated. In general, only very few clones were positive for reactivity to CVB, but ultimately it was not possible to generate a stable anti-CVB clone. However, specific peptide antibodies were generated, which recognized either GAD65, GAD67, or both.

Precautions when generating peptide antibodies are necessary, e.g., peptide antibodies often yield low titers, as peptides themselves are not immunogenic, and hence a proper carrier for presentation is necessary [35,36,37]. Moreover, it is crucial that the antibodies are selected in the assay that they are intended to be used in both in relation to immunoassay but to the final target as well. In the current studies, peptides were selected towards N-terminally biotinylated peptides. As a consequence, five out of six antibodies had a direct dependency of the free C-terminal carboxylic acid in combination with a seemingly random peptide backbone. The latter was confirmed when conducting a competitive inhibition assay, where antibody reactivity was inhibited with a random peptide from Epstein–Barr nuclear antigen 2, which has no sequence homology to GAD.

Only SSI-HYB 387-01 was specific for a target located within the immunized peptide, the RYKYFP sequence. This peptide is located in the N-terminal of the homologous PEVKEK sequence found in GAD65 and CVB P2C. In this region of GAD65, the sequence RFKMFP is found, and thus, although four out of six amino acids were identical in the two isoforms, no reactivity was observed to GAD65. Similarly, two amino acids of the SSI-HYB 387-01 epitope were located in the same position in P2C as well (KVKILP). These findings are in accordance with current literature, describing that antibodies most often are very specific for their generated targets unless the antibodies have a high degree of backbone dependency for antibody reactivity [38,39].

Analysis of the reactivity of SSI-HYB 387-01 to full-length GAD67 in Western blotting illustrated that the antibody only recognized the protein under reduced conditions. The question remains as to whether the antibody fails to recognize the native GAD67 monomer, as the protein only was found as a dimer and trimer under non-reduced conditions.

Although no antibody was generated that targeted all three proteins, the current findings may provide a clue about the antigenicity of the three peptides used for immunization. For this purpose, T1D sera were tested for reactivity to the three peptides. However, no reactivity was found to the three peptides, although antibody reactivity has previously been found in this region [33], and hence it remains to be determined as to whether differences in assay setup may explain these findings.

Collectively, the current findings may not support the theory of molecular mimicry between antibodies as a contributor to onset of T1D or SPS. 

The etiology of T1D and SPS onset still remains to be determined, and several factors have been proposed to be involved in disease onset. Recently, Th17 cells have been linked to several autoimmune diseases, e.g., arthritis, multiple sclerosis, and lupus. The proinflammatory actions of Th17 can be beneficial to the host during infection; however, uncontrolled activation has been proposed to be linked to the onset of autoimmune diseases and chronic conditions [40]. Thus, it may be hypothesized that uncontrolled activation of Th17 or other T cells by GAD isoforms or CVB presented on MHC II or I may contribute to initiate an unwanted cross-reactive immune response. This remains to be elaborated.

## 4. Materials and Methods

### 4.1. Reagents

Two-chloro-trityl-resins were from Novabiochem (Darmstadt, Germany), and 1-hydroxy-7-aza-benzotriazole (HOAt) was from GL Biochem (Shanghai Ltd, China). N,N’-Diisopropylcarbodiimide (DIC), piperidine (PIP), *N*-methyl-2-pyrrolidone (NMP), 9-fluorenylmethoxycarbonyl (Fmoc)-protected amino acids, Wang resin, and Oxyma Pure were from Iris Biotech GmbH (Marktredwitz, Germany). Dichloromethane (DCM), *N,N*-diisopropylethylamine (DIPEA), hexafluorophosphate azabenzotriazole tetramethyl uranium (HATU), 4-dimethylaminopyridine (DMAP), trifluoroacetic acid (TFA), para-nitrophenyphosphate (*p*NPP), ether, and alkaline phosphatase (AP)-conjugated goat anti-mouse IgG were from Sigma (Steinheim, Germany). Triisopropylsilane (TIS) was from Fluka Chemie AG (Buchs, Switzerland). Fmoc-L-amino acids were from Novabiochem (Darmstadt, Germany). Acetonitril was from Carlo Erba (Milano, Italia). Tris-Tween-NaCl (TTN) buffer (0.05 M Tris, 0.3 M NaCl, 1% Tween 20; pH 7.4), AP substrate buffer (1 M diethanolamine, 0.5 mM MgCl_2_; pH 9.8), and carbonate buffer (0.05 M sodium carbonate; pH 9.6) were from Statens Serum Institut (Copenhagen, Denmark). Full-length GAD and CVB peptides (Figure 1) were kindly donated by Schafer-N (Lyngby, Denmark). Resin-bound peptides used for epitope identification of SSI-HYB 387-01 were purchased from Schafer-N (Lyngby, Denmark). Free peptides used for epitope characterization were synthesized using Fmoc-based solid-phase peptide synthesis (SPPS), as described in Section 4.3. Peptides used for immunization were synthesized by microwave-assisted SPPS as described in Section 4.4 [41,42]. Recombinant GAD67 was from Baltymas (Vilnius, Lithuania). Human sera (*n* = 32) from T1D patients were from Steno Diabetes Center, Denmark. Samples were obtained from children with T1D in the age of 0-18 years. The samples were used anonymously. Healthy control sera (*n* = 10) were obtained from Statens Serum Institut and used anonymously.

### 4.2. Peptides

Peptides from human GAD65, human GAD67, and CVB4 P2C were used for immunization (Figure 1). Peptides used for antibody characterization are listed in Appendix A.

### 4.3. Solid-Phase Peptide Synthesis of Free Peptides

Free peptides were synthesized using standard SPPS protocols [43]. Tentagel S RAM resin (50 mg, loading: 0.24 mmol/g) was used for peptide synthesis. Fmoc-protected amino acids and coupling reagents (HOAt and DIC) were used in twofold excess. Coupling was performed for 2 h at room temperature (RT). The Fmoc group was removed using 20% *v/v* PIP in NMP for 20 min. Peptides and side-chain protecting groups were cleaved from the resin using 94% TFA, 1% TIS, 2.5% MilliQ water, and 2.5% DTT for 4 h at RT. Finally, TFA was removed by evaporation, and free peptides were precipitated in ether, redissolved in 10% aqueous acetic acid, and lyophilized. Following peptide lyophilization, the peptides were characterized using reverse phase ultra-high-performance liquid chromatography and electro-spray ionization mass spectrometry (Ultimate 3000—Thermo Fisher Scientific, Waltman, MA, USA).

### 4.4. Microwave-Assisted Solid-Phase Peptide Synthesis

GAD65, GAD67, and CVB peptides used for immunization were synthesized by automated microwave-assisted SPPS using the “Liberty Blue” by CEM corporation (Matthews, NC, USA) [42]. Pre-loaded Wang resins were used for synthesis. Oxyma Pure and DIC were utilized for activation, and a single coupling for each amino acid was performed. Cleavage of the peptides, along with removal of side-chain protecting groups, was conducted using 95% TFA, 2.5% MilliQ water, and 2.5% TIS. The resins were treated with the cleavage mixture under magnetic stirring for 90 min, whereafter the solution containing the cleaved peptide was filtered and treated with isopropyl ether at 4 °C to precipitate the crude peptide. The filtered solution was centrifuged at 3500 rpm (2 × 5 min), whereafter the supernatant was discarded. The pellet was vacuum-dried and dissolved in 50:50 ACN/MilliQ water. Following peptide lyophilization, the peptides were characterized using reverse phase ultra-high-performance liquid chromatography and electro-spray ionization mass spectrometry (Ultimate 3000—Thermo Fisher Scientific, Waltman, MA, USA).

### 4.5. Generation of Mouse GAD Antibodies

Peptide antibodies were generated using traditional protocols [35,36]. Synthesized peptides were pre-coupled to KLH before immunization. Three groups of four mice (A, B, C) were immunized with GAD65, GAD67, and CVB4 P2C peptides, respectively. Following immunization, mice were administered injections for 6–8 times in total. Mice bleeds were collected and tested for antibody reactivity. Following immunization, spleen cells were harvested and fused with a standard hybridoma cell line (X63.Ag8.653) using conventional technology [35,36]. Selection of reactive clones was conducted using N-terminal biotinylated peptides.

### 4.6. Testing of Antibody Reactivity to Biotinylated Peptides by Enzyme-Linked Immunosorbent Assay

Biotinylated peptides (1 µg/mL) were incubated in microtiter plates (2 h at RT) pre-coated with streptavidin (ON at 4 °C in carbonate buffer). Wells were blocked with TTN for 30 min at RF. Between blocking and incubation layers, the wells were washed with TTN 3 × 1 min on a shaking table. Wells incubated with patient serum (1:100), culture supernatant (1:10), secondary antibody (goat anti-mouse IgG (1:2000), and goat anti-human AP IgG (1:2000)) for 1 h at RT. Bound antibodies were detected by adding AP substrate buffer along with *p*NPP (100 µL, 1 mg/mL) to each well and reading at A_405–650_ on a microtiter plate reader after an appropriate color development.

### 4.7. Testing of Antibody Reactivity to Resin-Bound Peptides by Modified Enzyme-Linked Immunosorbent Assay

Resin-bound GAD/CVB peptides truncated systematically from the N-terminal or C-terminal (see Appendix A) were added to a 96-well multiscreen filter plate (Millipore, Copenhagen, Denmark) and rinsed with TTN buffer. All incubations with antibodies diluted in TTN (1:10 for cell culture supernatants, 1:1000 for purified antibodies) were carried out for 1 h at RT followed by three washes in TTN buffer. Resin beads were washed with TTN buffer using a multiscreen vacuum manifold. AP-conjugated goat anti-mouse IgG was used as primary antibody. Bound antibodies were quantified using *p*NPP (1 mg/mL) diluted in AP substrate buffer. Finally, the buffer was transferred to a Maxisorp microtiter plate (Nunc, Roskilde, Denmark), and the absorbance was measured at 405 nm, with background subtraction at 650 nm, on a Thermo max microtiter plate reader (Molecular Devices, San Jose, CA, USA).

### 4.8. Competitive Inhibition Assay for Final Epitope Identification of SSI-HYB 387-01

The wells of microtiter plates were coated with free GAD67 peptide (1 µg/mL) in 20 mM Tris (pH 7.5) for 2 h at RT. SSI-HYB 387-01 was diluted (1:10) in TTN and incubated with relevant peptides (100 µg/mL) for 1.5 h (end-over-end) at RT. Next, the GAD67-coated microtiter plate was rinsed with TTN for 3 x 1 min, and the SSI-HYB 387-01-peptide solution was added to the microtiter plate and incubated for 1 h at RT. The wells were rinsed using TTN for 3 × 1 min, whereafter AP-conjugated goat anti-mouse IgG (1:2000) was added to the plate and incubated for 1 h at RT. Following rinsing of the microtiter plate, as just described, AP substrate buffer (1 mg/mL) was added to the wells, and bound antibodies were determined by measuring the absorbance at 405–650 nm on a microtiter plate reader.

### 4.9. Competitive Inhibition Assay Determination of Antibody Reactivity

The wells of microtiter plates were coated with streptavidin (1 µg/mL) ON, followed by three washes of TTN 3 × 1 min and incubation with B-GAD67 or B-GAD65 peptide (1 µg/mL) in carbonate buffer for 2 h at RT. SSI-HYB 387-01 was diluted (1:10) in TTN and incubated with relevant peptides (500 µg/mL) (GAD67/GAD65, CVB, random control peptide) for 1 at RT. Next, the coated microtiter plate was rinsed with TTN for 3 × 1 min, and the SSI-HYB 386-01/386-02/387-02-peptide solution was added to the microtiter plate and incubated for 1 h at RT. The wells were rinsed using TTN for 3 × 1 min, whereafter AP-conjugated goat anti-mouse IgG (1:2000) was added to the plated and incubated for 1 h at RT. Following rinsing of the microtiter plate, as just described, AP substrate buffer (1 mg/mL) was added to the wells, and bound antibodies were determined by measuring the absorbance at 405–650 nm on a microtiter plate reader.

### 4.10. SDS-PAGE and Western Blotting

A total of 10 µg recombinant GAD67 was diluted 1:2 in reducing and (non-)reducing sample buffer (SB) (+/− dithiothreitol (DTT)) and incubated for 2 min at 95 °C. Twenty microliters of each sample was loaded to wells of a 4–20% Tris glycine gel (Novex Life Technologies, CA, USA) and run for 2 h at 50 V and 250 mA and later 30 min at 100 V and 250 mA using Tris glycine SDS running buffer. Gels were stained overnight at 4 °C with Coomassie Brilliant Blue and washed with milli-Q water 5 times until bands were visualized.

For Western immune-blotting, the procedure was repeated as described above. After electrophoresis, gels were blotted onto a PVDF membrane using an iBlotter (Invitrogen by Thermo Fisher, Waltman, MA, USA). The membranes were blocked overnight in TTN buffer and mounted in a miniblotting device. GAD67 antibody was diluted 1:200 in TTN, added to each well, and incubated for 1 h at RT. Washes were performed for 2 × 5 min, first in wells, and afterwards in a plastic dish with TTN. AP-conjugated goat anti-mouse IgG was diluted 1:1000, incubated for 1 hour on the membrane, followed by washing 3 × 5 min. Finally, AP substrate (BCIP 0.5 mg/mL, NTB 0.3 mg/mL, Sigma, St. Louis, MO, USA) was added and incubated. The reaction was stopped by washing the membrane in milliQ water, followed by drying on filter paper.

### 4.11. Data Analysis

Statistical analyses presented in Figure 2 were generated using Prism 5 software (Graphpad, San Diego, CA, USA). Statistical significance was assessed by nonparametric unpaired two-tailed Mann–Whitney U-test. Significant differences are indicated by *: *p* < 0.05, **: *p* < 0.01, ***: *p* < 0.001.

## Figures and Tables

**Figure 1 ijms-23-04424-f001:**
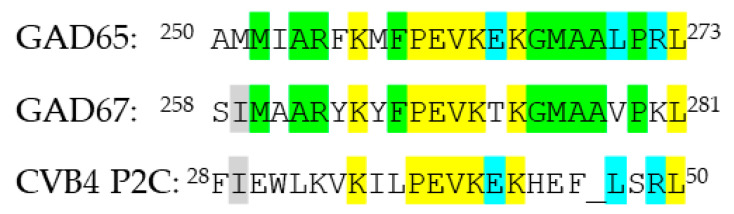
Sequence homology of glutamic acid decarboxylase (GAD)65, GAD67, and coxsackievirus (CVB)4 P2C peptides. Highlight in yellow represents amino acids found in all three peptides. Highlight in green represents overlap between GAD65 and GAD67. Highlight in blue represents overlap between GAD65 and CVB. Highlight in grey represents overlap between GAD67 and CVB. Similarity among GAD65, GAD67, and CVB4 P2C peptides are: GAD65/GAD67 = 67% overlap (16 identical amino acids), GAD65/CVB4 P2C = 42% overlap (10 amino acids), and GAD67/CVB4 P2C = 29% overlap (7 amino acids). The three peptides were used for generation of peptide antibodies.

**Figure 2 ijms-23-04424-f002:**
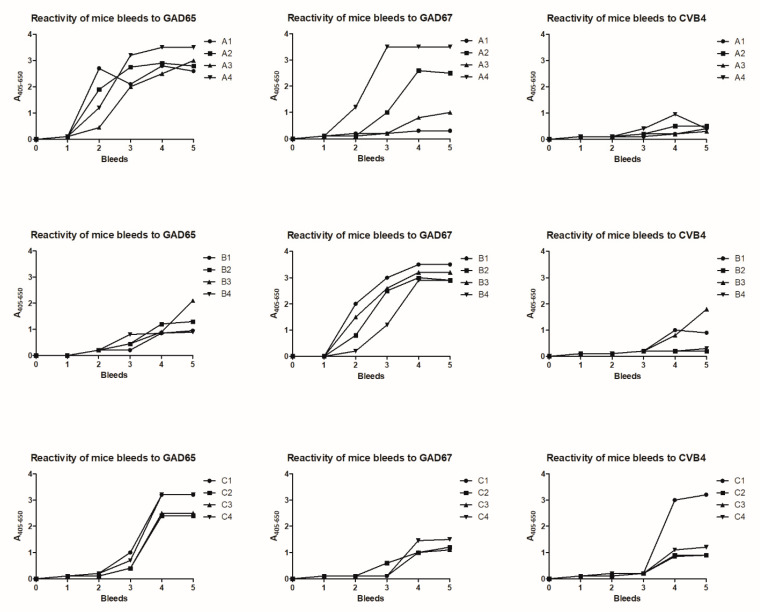
Reactivity of mice bleeds to glutamic acid decarboxylase (GAD)65, GAD67, and coxsackievirus (CVB) peptides in enzyme-linked immunosorbent assay. Biotinylated peptides were coated to pre-coated streptavidin microtiter plates, whereafter samples were added and antibody reactivity was determined. Bleed 0 was obtained prior to immunization, whereas the remaining were obtained after immunization. Statistically significant elevated antibody reactivity was found to the bleeds from the collected mice bleeds from bleed 5 compared to bleed 0.

**Figure 3 ijms-23-04424-f003:**
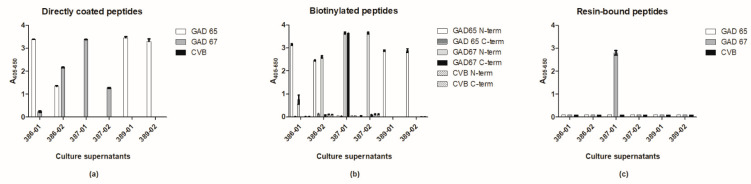
Antibody reactivity to glutamic acid decarboxylase (GAD) and coxsackievirus (CVB) peptides in different enzyme-linked immunosorbent assay (ELISA) setups. (**a**) Antibody reactivity to peptides coated directly to the solid surface of microtiter plates. (**b**) Antibody reactivity to biotinylated peptides tested in pre-coated streptavidin microtiter plates. (**c**) Antibody reactivity to resin-bound peptides analyzed by modified ELISA, where reactivity was determined though interaction to resin-linked peptides in membrane-containing microtiter plates, as described in the Section 4.

**Figure 4 ijms-23-04424-f004:**
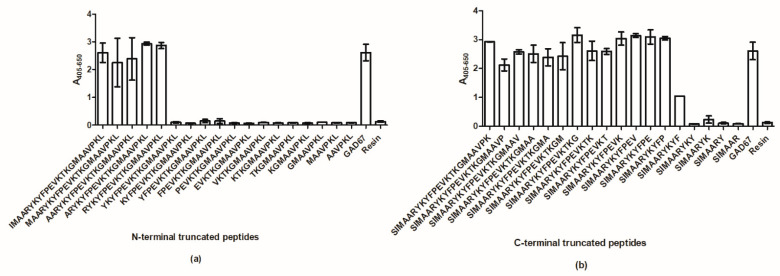
Reactivity of SSI-HYB 387-01 to N- and C-terminal-truncated resin-bound peptides analyzed by modified enzyme-linked immunosorbent assay. (**a**) Reactivity to N-terminal-truncated peptides. (**b**) Reactivity to C-terminal-truncated peptides. Full-length glutamic acid decarboxylase (GAD)67 peptide was used as positive control. Resin was used for background determination.

**Figure 5 ijms-23-04424-f005:**
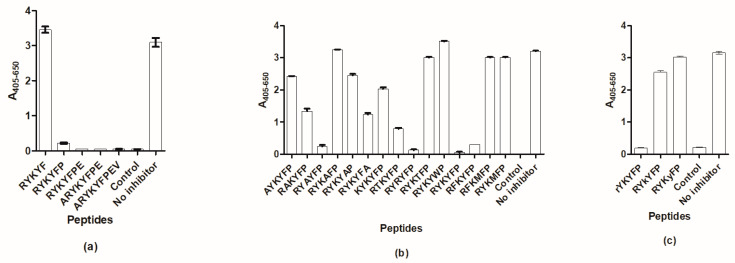
Epitope identification of SSI-HYB 387-01 analyzed by competitive inhibition assays. (**a**) Antibody reactivity to epitope variants. (**b**) Antibody reactivity to Ala- and functionality-substituted peptides. (**c**) Antibody reactivity to D-amino acid-substituted peptides. Glutamic acid decarboxylase (GAD)67 full-length peptide was used as a positive control.

**Figure 6 ijms-23-04424-f006:**
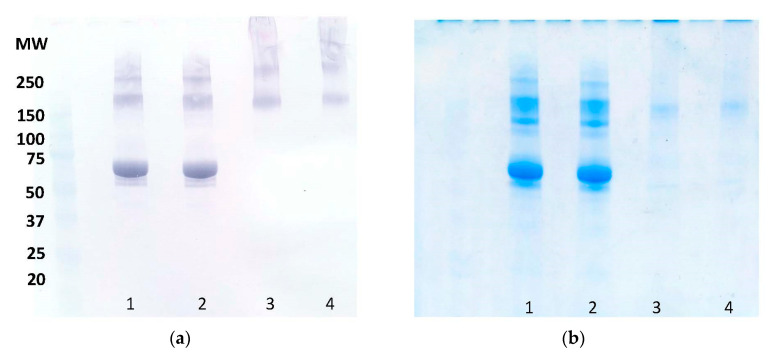
Immunostaining of SSI-HYB 387-01 reactivity. (**a**) Western blotting of SSI-HYB 387-01 reactivity to full-length glutamic acid decarboxylase (GAD)67 protein. (**b**) Coomassie Brilliant Blue staining of recombinant GAD67. Lane 1 + 2: reduced recombinant GAD67, lane 3 + 4: non-reduced recombinant GAD67.

**Figure 7 ijms-23-04424-f007:**
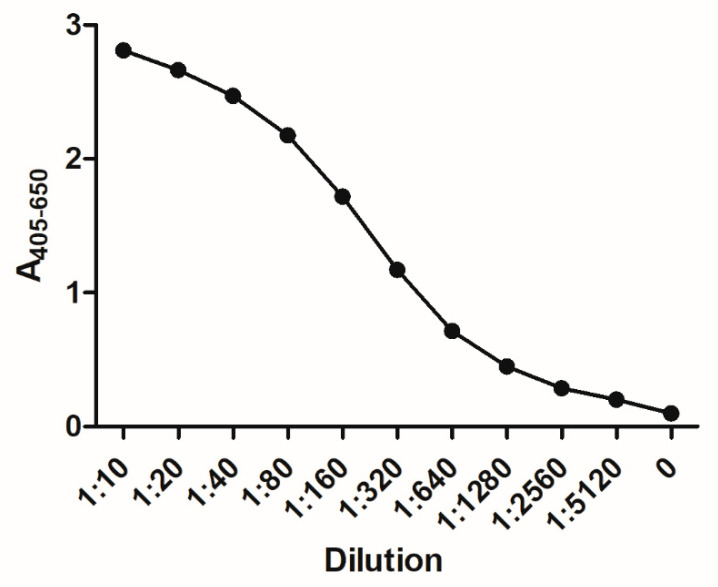
Antibody titration of SSI-HYB 387-01 to glutamic acid decarboxylase (GAD)67 analyzed by enzyme-linked immunosorbent assays.

**Figure 8 ijms-23-04424-f008:**
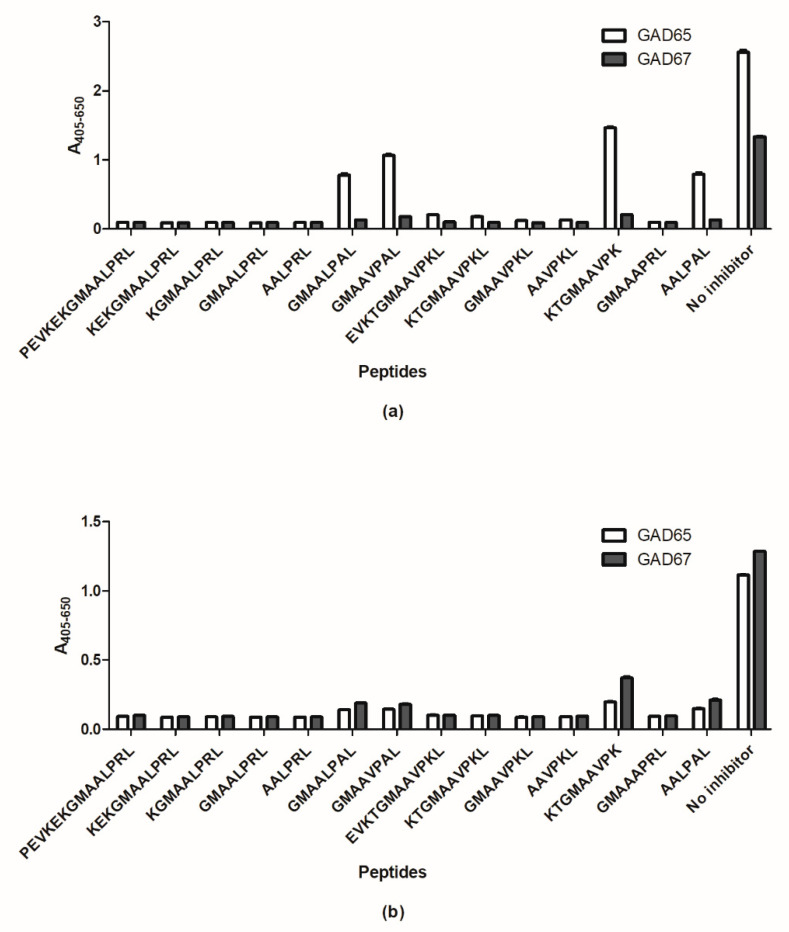
Identification of epitopes of SSI-HYB 386-01 and -02 analyzed by competitive inhibition assay. (**a**) Reactivity of SSI-HYB 386-01 to glutamic acid decarboxylase (GAD)65 and (GAD)67. (**b**) Reactivity of SSI-HYB 386-02 to GAD65 and GAD67.

**Figure 9 ijms-23-04424-f009:**
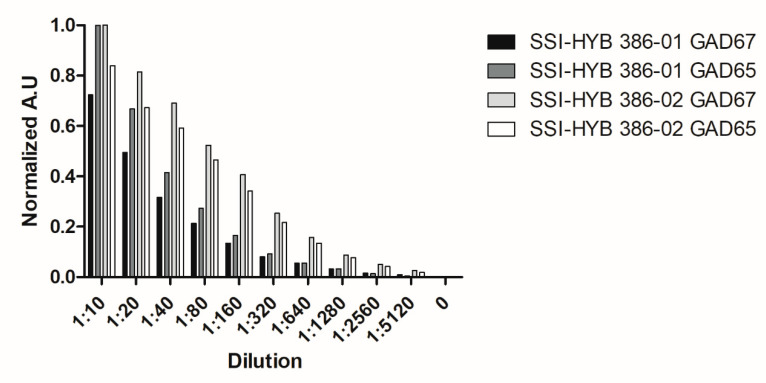
Reactivity of SSI-HYBs analyzed by enzyme-linked immunosorbent assay.

**Figure 10 ijms-23-04424-f010:**
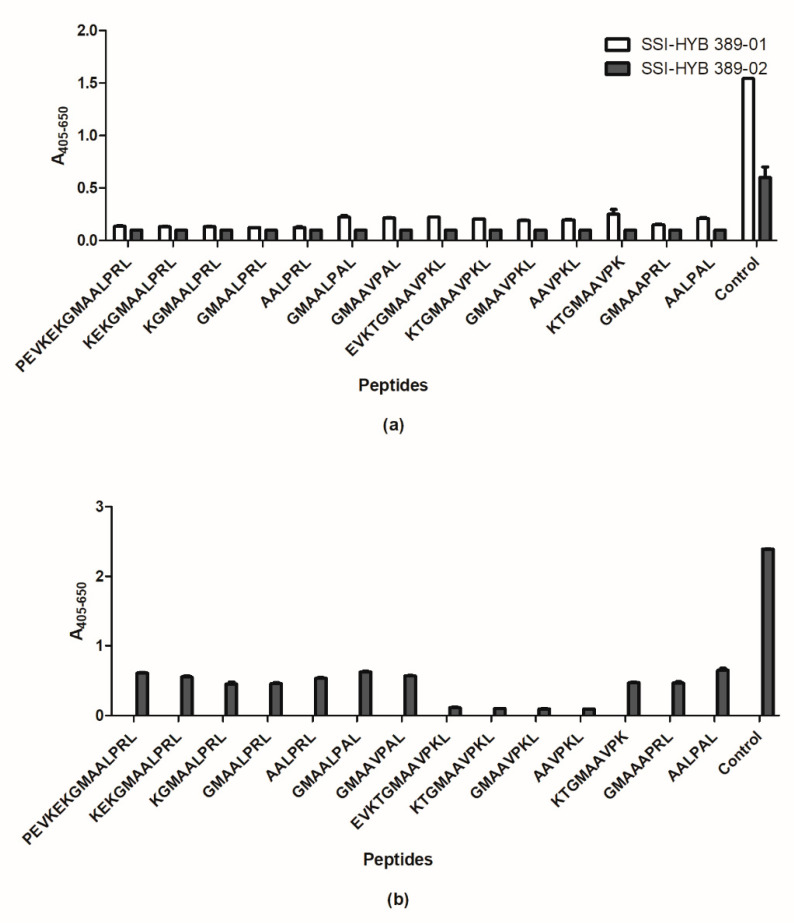
Reactivity of SSI-HYB 387-02 and SSI-HYB 389-01 and -02 to truncated and substituted glutamic acid decarboxylase (GAD) peptides analyzed by competitive inhibition assay. (**a**) Reactivity of SSI-HYB 389-01 and 389-02 to GAD65. (**b**) Reactivity of SSI-HYB 387-02 to GAD67.

**Figure 11 ijms-23-04424-f011:**
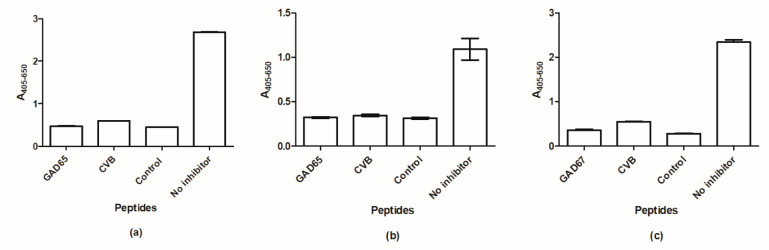
Reactivity of SSI-HYB 386-01, SSI-HYB 386-02, and SSI-HYB 387-02 analyzed by competitive inhibition assays. (**a**) Reactivity of SSI-HYB 386-01 to glutamic acid decarboxylase (GAD65). (**b**) Reactivity of SSI-HYB 386-02 to GAD65. (**c**) Reactivity of SSI-HYB 387-02 to GAD67. Inhibiting peptides; GAD65/67, CVB, control peptide (GQGRGRWRGRGRSKGRGRMH).

**Figure 12 ijms-23-04424-f012:**
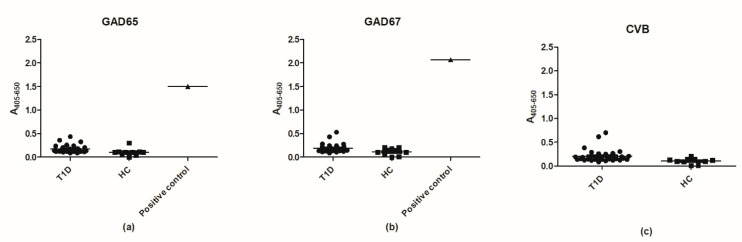
Reactivity of type 1 diabetes sera (T1D) (*n* = 32) and healthy donor (HD) sera (*n* = 10) to glutamic acid decarboxylase (GAD)65, GAD67, and coxsackievirus (CVB) peptides analyzed in enzyme-linked immunosorbent assay. (**a**) Reactivity of T1D and HC sera to the full-length GAD65 peptide. (**b**) Reactivity of T1D and HD sera to the full-length GAD67 peptide. (**c**) Reactivity of T1D sera and HD sera to the full-length CVB peptide.

**Table 1 ijms-23-04424-t001:** Antibody clones selected for final characterization.

Name	Immunogen	Clone Number	Subclass
SSI-HYB 386-01	GAD65	SSI-8F8	IgG2b/kappa
SSI-HYB 386-02	GAD67	SSI-13G5	IgG1/kappa
SSI-HYB 387-01	GAD67	SSI-15D9	IgG2a/kappa
SSI-HYB 387-02	GAD67	SSI-8D2	IgG1/kappa
SSI-HYB 389-01	GAD65	SSI-13B6-Y	IgG1/kappa
SSI-HYB 389-02	GAD65	SSI-1G9-A	IgG1/Kappa

## Data Availability

Not applicable.

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
