# Peer review of "Peptide Antibody Reactivity to Homologous Regions in Glutamate Decarboxylase Isoforms and Coxsackievirus B4 P2C"

_ijms, 2022, doi:10.3390/ijms23084424_

Round 1

Reviewer 1 Report

The paper is interesting and well written. I suggest only to discuss the potential capacity of Th17 to recognize glutamate decarboxylase and coxsachievirus antigens (see and add as reference paper by Murdaca et al concernoing Th17 in chroni cimmunemediated diseases)

Author Response

Reviewer,

Thank you for the constructive comments raised. Th17 and their potential role in disease onset has been added in the discussion together with a reference to the manuscript by Murdaca et al.

Sincerely

Gunnar Houen and Nicole Trier

Reviewer 2 Report

Trier’s et al. manuscript is devoted to the fundamental research of the peptide antibody reactivity to GADs and CVBs homologous regions. In general, the manuscript is well written and strongly discussed. To make it a bit clearer for the readers I suggest the following minor revision.

  1. Taking into account that the paper starts with Results but not with Experimental part, some introductory paragraph explaining the design of experiment is highly desirable. Also, I would recommend moving Figure 12 from Experimental part to Results to make further reading clearer.  
  1. Line 98.

What is KLH? Is it abbreviation or peptide sequence? I have not find any explanation of this. Please, precise.

  1. Figure 1.

Please, add statistics to these results.

  1. Title of Table 1. The title contains twice repeated word “selected”. Please rephrase the title.
  1. Line 310 refers to Figure 1. Check if it is correct. I did not find an information addressed in the text in the pointed Figure.
  2. Section 4.3. How were the peptides purified? HPLC? Conditions?
  3. A manuscript contains a lot of minor issues which should corrected. Please re-read the text carefully. Some examples, line 169 – alanine should be written with a small letter “a”; in line 437 volume unit is written as “L” (µg/mL) while in line 460 as “l” (mg/ml). µL in line 448 is written as ul. Please unify the way units are written; somewhere the spaces between words are absent (line 450 – 50V, 250mA; line 403 – 1µg/mL, etc.)

Author Response

Dear reviewer,

Thank you for the constructive comments raised by the reviewers. We have addressed all of the comments, which have improved the manuscript considerably.

Sincerely

Gunnar Houen and Nicole Trier

Reviewer 2:
Trier’s et al. manuscript is devoted to the fundamental research of the peptide antibody reactivity to GADs and CVBs homologous regions. In general, the manuscript is well written and strongly discussed. To make it a bit clearer for the readers I suggest the following minor revision.

1. Taking into account that the paper starts with Results but not with Experimental part, some introductory paragraph explaining the design of experiment is highly desirable. Also, I would recommend moving Figure 12 from Experimental part to Results to make further reading clearer.  

 Response: Amended as requested. The original figure 12 is now figure 1 and placed in the introduction section, clearly illustrating the cross-reactive regions between the three peptides. The experimental setups have been elaborated in the text and figure legends where appropriate.

2. Line 98. What is KLH? Is it abbreviation or peptide sequence? I have not find any explanation of this. Please, precise.

Response: The abbreviation KLH has been added to the text. It is short for keyhole limpet haemocyanin, which is an immunogenic protein.

3. Figure 1. Please, add statistics to these results.

 Response: Statistics have been added to the figure.

4. Title of Table 1. The title contains twice repeated word “selected”. Please rephrase the title.

Response: The title has been rephrased.

5. Line 310 refers to Figure 1. Check if it is correct. I did not find an information addressed in the text in the pointed Figure.

Response: The sentence has been rephrased.

6. Section 4.3. How were the peptides purified? HPLC? Conditions?

Response: The peptides were purified by HPLC using standard conditions and described elsewhere in the manuscript. This has been added to the paragraph.

7. A manuscript contains a lot of minor issues which should corrected. Please re-read the text carefully. Some examples, line 169 – alanine should be written with a small letter “a”; in line 437 volume unit is written as “L” (µg/mL) while in line 460 as “l” (mg/ml). µL in line 448 is written as ul. Please unify the way units are written; somewhere the spaces between words are absent (line 450 – 50V, 250mA; line 403 – 1µg/mL, etc.)

Response: Thank you for noticing! The minor mistakes have been corrected.